# Exploring Environmental and Entrepreneurial Antecedents of Social Performance in Spanish Sports Clubs: A Symmetric and Asymmetric Approach

**Paloma Escamilla-Fajardo**[ID]**, Juan Manuel Núñez-Pomar * and Ana María Gómez-Tafalla**

Department of Physical Education and Sports, Faculty of Physical Activity and Sport Sciences, University of Valencia, 46010 Valencia, Spain; paloma.escamilla@uv.es (P.E.-F.); Ana.Maria.Gomez@uv.es (A.M.G.-T.)
* Correspondence: juan.m.nunez@uv.es

**Abstract:** The social function of non-profit sports clubs is undeniable, so analyzing the factors that influence their performance is vital. The aim of this study is to understand the influence of entrepreneurial factors (entrepreneurial orientation) and environmental factors (dynamism, hostility and complexity) on social performance using a symmetrical (Hierarchical regression model) and asymmetrical (qualitative comparative analysis) approach. The social performance of this particular type of organization is of great importance in an environment where sustainability from a social point of view is increasingly on the agenda of governments, organizations and society in general. A total of 209 Spanish non-profit sports clubs were analyzed. The use of two complementary methodologies has made it possible to highlight the direct positive influence of entrepreneurial orientation (EO) on performance in terms of social impact performance (SIP) and social causes performance (SCP). Similarly, complexity and dynamism have a direct influence on both types of performance, while high levels of hostility in the environment are related to low levels of social performance in both areas. The analysis of the interaction of environmental factors and sport entrepreneurship on social performance in sports clubs has not been previously addressed. Therefore, this study provides new information to elaborate on practical management implications for directors and managers of non-profit sport clubs.

**Keywords:** entrepreneurial orientation; sport entrepreneurship; environmental; dynamism; complexity; social performance; sports clubs; sports organizations

---

## 1. Introduction

Physical activity and sport can have important health-related mental and physical benefits [1], as well as reducing the risk of serious diseases such as obesity and heart disease [2]. This is why European governments have focused their attention and efforts in recent decades on promoting physical activity and sport, from the "sport for all" policies in the 1960s and 1970s to the current sport policies that promote that as many people as possible can be physically active without feeling socially excluded [3]. Before European policies including Sports for All, sport was mainly an activity for a select group in society [4], however, this has evolved in a surprising way, and sport has become a major economic, social and cultural activity.

In this context, sports clubs have been able to attract the attention of academics and professionals from an entrepreneurial and social perspective [5], as in many developed countries they are the main provider of sports programs for society [6,7]. In Spain, there were 67,512 registered sports clubs and 3,866,867 federal licenses in 2018 [8]. However, it should be noted that not all people who engage in physical activity and sport in a sports club are members of a federation, so sports clubs are attended by a higher percentage of

society. Unlike in the past, new groups are finding their place in sports clubs: older people and women [4], refugees [9], people with disabilities [10] and people at risk of poverty [11].

As can be seen, and due to a more extensive and heterogeneous demand, sports clubs have been forced to improve their structure, adjust the services that they offer and achieve greater professionalization to meet the expectations and needs of their members and users [12]. This type of non-profit sports entity is characterized by voluntarism [13], the high mobility of the sportsmen and women who make it up and its solidarity. Furthermore, due to their status as non-profit organizations that cannot produce profits at the end of the year, and their social responsibility, sports clubs focus on sports and social aspects above any financial or economic objective [14], maintaining an outstanding socializing role and a broad social function.

Social objectives are closely related to the promotion of sport in the community [15], which gives them vital importance in social development [16]. However, despite government efforts to bring sport closer to the population, several researchers have found that people with lower socioeconomic status are less likely to engage in sport [17], just as immigrants or people with disabilities encounter major barriers on the road to sports participation [18]. Hence, one of the greatest social and governmental concerns is to facilitate access to physical activity and sport for all people, by providing public subsidies to non-profit sports clubs which carry out important social work [15]. This social work is related to providing access to physical activity and sport to the largest possible number of people, improving their general well-being [19]. Consequently, the social purpose of sports clubs can be understood from a double perspective: (i) "social impact performance" is understood as the promotion of sport in a broad sense, with the aim of reaching the greatest possible number of members of a community through the development and implementation of a high number of programs and services. This would be an approach focused on the quantitative impact that the organization has on its environment, and would undoubtedly be a relevant performance indicator; while (ii) "social causes performance" is understood as the aim of bringing sport closer to groups with special needs or at risk of exclusion through social programs and solidarity actions.

Currently, despite a wide and heterogeneous demand for sports services, clubs have to compete with private sports organizations and other sports clubs offering similar services [20,21]. In fact, sports clubs carry out their professional, sporting and social activities under similar environmental conditions as other private sports organizations do at the political, social, sporting and cultural levels. Similarly, sports clubs are under pressure from public institutions on their sporting and social activity [11]. Therefore, this type of entity operating in a complex, dynamic and changing environment is forced to know and assimilate the information generated by the environment in order to offer a quick and adequate response [5]. This environment forces the organization to adopt entrepreneurial strategies to achieve and maintain the success of the organization [22].

The interest in analyzing the role of entrepreneurial orientation (EO) in the performance of sports organizations has grown exponentially in the last decade [22], since EO is considered as one of the most studied variables in relation to organizational entrepreneurship [23]. In this context, there is literature that analyzes the predictive capacity of EO in the performance of sports clubs [5,24]. However, despite the fact that the discovery of new business possibilities or opportunities depends largely on previous knowledge of the rest of the organizations (competition) and the ability to analyze and explore the environment surrounding the organization [25], the literature is scarce on studies that jointly analyze the influence of environmental factors and entrepreneurial factors on the social performance of sports clubs. In non-profit sports clubs, the social function stands out ahead of the economic objective, hence social performance is vital in this type of organization. However, there is scarce literature that focuses attention on the impact of entrepreneurship on social performance, and this is non-existent in the sports sector. Therefore, this study can provide relevant information to this gap in knowledge.

Covin and Slevin [26] studied the influence of entrepreneurship on economic performance in turbulent environments, while Lumpkin and Dess [27] already analyzed the influence of munificence, complexity and dynamism on the EO-Firm performance relationship. Therefore, there are studies

that consider the environment in the interesting entrepreneurship-performance relationship, however, we find a wide gap in the literature in the field of sports. There are theoretical studies that consider the environment in the sports field [4,28], however, empirical works are needed to understand the role that dynamism, complexity and hostility, along with entrepreneurship in the performance of non-profit sports organization. Furthermore, the relationship between sport entrepreneurship and social performance has not been previously addressed, and this article may provide new and useful information for both the non-profit sector and the general sports field.

Therefore, the main objective of this study is to analyze the influence of EO and factors from the organization's environment (dynamism, hostility and complexity) as background on the social performance of Spanish sports clubs through symmetric (linear regressions) and non-symmetric (fsQCA) methodologies. One of the main contributions of the present study is the analysis of entrepreneurial and environmental factors on the social performance of the sports association network in Spain. This analysis has not been previously carried out in the sports industry, and even less so if we take into account non-profit sports organizations, so it can provide valuable information for the field of action.

The structure followed in the paper would be as follows: following this introduction, Section 2 reviews and discusses the existing literature and the presentation of research proposals. Section 3 explains the part of the methodology that includes the description of the sample, the instrument used, the procedure and the data analysis. Section 4 presents the results obtained taking into account the two methodologies used. Section 5 discusses the results in relation to the previous literature. Finally, Section 6 presents the main conclusions and practical implications, along with the limitations of the study.

## 2. Literature Review

In recent years, interest in analyzing entrepreneurial orientation (EO) has increased considerably [23,24], as it is considered to be one of the most important constructs within the entrepreneurial ecosystem [29]. In the current context of competitiveness and constant change, organizations are forced to adopt entrepreneurial strategies to achieve and maintain the success of their organization. In this sense, many academics and practitioners have exposed the relationship between EO and overall organizational performance [29–31].

EO can be seen as a strategic orientation that has an important role in exploring the opportunities and threats of the environment and competition, and in exploiting existing resources and generating new processes and services to maintain a strong market position. EO reflects how an entity works, while entrepreneurial activities affect characteristics, decisions, actions, processes and the performance of the organization [27]. EO is traditionally a multidimensional construct that encompasses three main dimensions: risk-taking, innovativeness and proactiveness [32].

Risk taking is understood as the level of risk that the management of the entities is willing to assume, exposing notably own resources with the projection of achieving benefits as a return. According to Lumpkin and Dess [27], it involves decisions and actions that use significant resources in a way that compromises them. Proactivity is related to the anticipation of changes in the environment. This requires detailed knowledge of the sector and of competitors in order to carry out actions that provide a competitive advantage to the organization [5]. Finally, innovation is the individual capacity of the organization to create and experiment with new ideas, through the use of creative processes and the incorporation of new products and services [33]. For some authors, innovation is the most important part of which EO is composed [34], but all dimensions are vitally important in the formation of the construct.

In the field of sports, in recent years "sport entrepreneurship" has emerged as a paradigm within entrepreneurship [35]. Sport entrepreneurship can be understood as the attitude of sports organizations committed to actively seeking opportunities [35]. "Sports entrepreneurship can involve a range of activities from experimentation of new ideas to the testing of different production methods" [36] (p. 10).

In recent years, it has attracted the attention of academics and professionals [37], although there are still more theoretical contributions and there is a gap in the literature of empirical works that analyze this construct [38].

The analysis of empirical and theoretical studies on the background of organizational performance shows that there is a variety of approaches to the subject. Nevertheless, there are numerous studies that show that high levels of entrepreneurial orientation are a reliable guarantee for achieving high levels of effectiveness and performance of the entity [39,40]. Generally, the relationship between EO and performance is related to the economic performance of a sports organization [24,41]. However, sports clubs, which are non-profit organizations, have social performance as their main objective [42]. Therefore, analyzing the influence of sport entrepreneurship on the social performance of sports clubs can provide valuable information for both the sports and non-profit sectors.

Sports entrepreneurship can have an improvement in the social impact of certain communities, having been approached from a phenomenological approach by Hemme et al. [43] in their study. Sports organizations have a predominantly social mission [44]. Sports organizations are characterized by their entrepreneurial nature, as they constantly adapt to changing societal needs, even as their products or services have a high social impact. Therefore, due to the high social impact and effects on communities that non-profit sports organizations have, it is necessary to explore the influence of entrepreneurial factors on social performance.

The social performance of sports clubs analyzed in this study can be understood from two similar, but different, perspectives: (i) social impact performance; and (ii) social causes performance. Social impact performance (SIP) is aligned with a basic, and generally common, objective of sports clubs to achieve the greatest possible impact on the environment in terms of number of members, programs developed or knowledge and awareness in the community. The capacity to generate services and products that offer the possibility of access to physical activity and sport to the population is aligned with their role as a complement to the work of government organizations in the community to promote sport, which gives them an important social function. On the other hand, social causes performance (SCP) emerges as a specific social response that the organization makes towards minority sectors or at risk of exclusion. Sport can facilitate integration and social inclusion [4], therefore, facilitating access to physical activity and sport for groups at risk of exclusion is one of the main social objectives of sports clubs. Moreover, the benefit can be mutual, as the club promotes the practice of sport that brings benefits to people at risk of exclusion, while these people help to maintain and sustain the sports clubs. This function could be perfectly aligned with the organizations' Corporate Social Responsibility policies, and let us not forget either that social responsibility is an element of legitimacy that sports organizations have from society [45] just as sporting events do [46].

Although performance in non-profit organizations needs even more theoretical and empirical study [47], the performance of organizations can be measured through subjective perception measures [48]. This study uses subjective questions to technicians or directly to the sports club that has a global vision of the organization. Based on the previous studies and the knowledge of the peculiarities of the sports sector, the main hypothesis would be the following:

**Hypothesis 1 (H1).** *Entrepreneurial orientation has a positive influence on SIP and SCP in sports clubs.*

The external environment of an organization and the influence on its performance has been widely studied in the theoretical and empirical organizational literature as having a high influence on the final product [49]. The external environment surrounding an organization has long been linked to its overall performance [50]. However, despite extensive study by different authors such as Thompson [51] or Miller and Friesen [52], full understanding of the environment-organization relationship is difficult due to its complexity. In the sports sector, sports clubs are not independent entities, but are included in a social, organizational and cultural environment. These organizations seem to be very dependent on the context or environment in which they operate, mainly in terms of information, resources and the

existence of opportunities that can be explored and exploited [53]. However, despite the importance of including environmental factors in organizational and entrepreneurial studies, there are few occasions when the dynamic and changing environment in which sports clubs operate has been assessed [4,11].

The relationship between environment and performance is complex because of the factors involved [54]. First, the analysis of the organizational environment can be done from two different perspectives (i) considering the different dimensions it encompasses (these being different depending on the authors analyzing it); and (ii) taking into account the influence of normative, cognitive and regulatory factors [54]. Most studies analyze the environment based on its most important dimensions [45,52,55]. These dimensions may be different depending on the authors who study the organization's environment. According to a review by Khandwalla [56], there are three environmental properties that most affect organizations: (i) environmental uncertainty; (ii) environmental hostility; and (iii) complexity, while Covin and Slevin [26] studied the environment formed by a single dimension, hostility. In the same vein, years later, Rosenbusch et al. [49], identified four key dimensions in the environment of organizations: munificence, dynamism, hostility and complexity, so that more detailed information can be given on what and how it affects performance. This last approach to the organizational environment has been widely shared in the sporting arena [57]; however, in this study we will build on the dimensions addressed by Miller et al. [58] and later by Zahra [59] and Frese et al. [60]: dynamism, hostility and complexity as the main factors in the external environment of organizations.

Dynamism is related to the unpredictability and uncertainty of future events and changes in the market [44,49]. According to Miller and Friesen [52], uncertainty can manifest itself in many different ways such as changes in user needs, variations in the behavior of competing organizations or the emergence of new technologies that can be adopted. More dynamic environments can be associated with high unpredictability, both in the demands of users or customers themselves and in the capabilities of competitors, so that rates of change are high and market trends easily alterable [61]. The sports sector is characterized by operating in a dynamic environment as new sports disciplines are constantly emerging and the demands and expectations of users and athletes are constantly changing.

The dynamic environment can have a significant impact on an organization's own capabilities and knowledge [62]. According to Zahra [59] and Miller and Toulouse [63], a dynamic environment can lead to the search for opportunities to achieve success and is closely related to the performance of the organization. In this context, rivalry between competitors can be intensified, leading to the creation of opportunities. In contrast, Wiklund and Shepherd [64] argued that a dynamic environment could have a negative influence on the organization's own performance. These data should be treated with prudence, since dynamic environments can be related to innovative actions and strategies by organizations, and thus improve organizational performance. However, dynamic environments can change into turbulent social environments [65] and have a negative impact on the attitude of organizations [66]. Based on previous literature and knowledge of the sports sector, the hypothesis formulated would be as follows:

**Hypothesis 2 (H2).** *The dynamism of the environment positively influences SIP and SCP in sports clubs.*

Heterogeneity refers to both diversity and the amount of skills, knowledge, resources and information needed to act successfully in a given context [67]. According to Zahra [59] (p. 264) "heterogeneity indicates the existence of multiple segments, with varied characteristics and needs, that are being served by the firm". This can be the result of complexity in organizations, that is, the lack of similarity between the main elements that surround an organization, or by the creation and offer of personalized and complex products or services [49]. In this study, heterogeneity is treated from the level of complexity that a sector may show at a given time. In this way, complexity is a variable that changes in the period in which it is analyzed and can be perceived in a very different way depending on the organization.

In this sense, the sports industry has a high degree of complexity, due to the large number of new sports disciplines, products or services demanded by users. The sports sector tries at all times to be innovative and proactive in meeting the expectations and needs of customers, athletes and users who request their services. These special characteristics give the sector an almost inherent complexity. As with dynamism, the complexity of the environment can lead the organization to explore a greater number of entrepreneurial opportunities and also to explore different processes and strategies to encompass and meet the needs of the largest possible population ratio. According to Miller and Friesen [52] organizations operating in complex environments receive different ideas from competitors and are therefore more likely to develop innovations. Several studies have analyzed the impact of the complexity of the environment on the organization in other sectors of activity; however, despite its undeniable importance, the analysis of the environment in the sports sector has not been widely studied [28,68], and even less so in the associative sports sector (sports clubs). So, the hypothesis raised from the existing literature is the following:

**Hypothesis 3 (H3).** *The complexity of the environment positively influences SIP and SCP of sports clubs.*

Finally, hostility is considered an unfavorable condition since it involves competition with other organizations for opportunities and scarce resources in this environment [26]. According to Khandwalla [56], it is understood as the degree of threat that the intensity or vigor of competition can pose to an organization. It involves concentration, legal, political and economic aspects that diminish profit margins and reduce strategic options, making it necessary to have a well-defined strategic discipline [52]. Hostile environments are identified with intense competition, scarcity of opportunities and many unexpected changes. March and Simon [69], were the first to name this concept (hostile environment), although they already developed the idea that a hostile environment forces quick reactions from the organization [67]. However, there is no consensus in the previous literature on how this background of environment affects the performance of the organization, and even less so in the sports sector.

The role of hostility in the social performance of sports clubs has not been addressed to date, however, the ways in which the role or influence of hostility on organizational performance is addressed are diverse. There are authors who expose a direct or moderating positive relationship of hostility to performance [26,70], while other authors identify the relationship between hostility and organizational performance as negative [49,71]. Therefore, in the present study, following the previous literature, the research hypothesis generated between a hostile environment and social performance is as follows:

**Hypothesis 4 (H4).** *A hostile environment has a negative influence on SIP and SCP of the sports clubs.*

The objectives of the research can be seen reflected in the conceptual model proposed by the authors in Figure 1. The objective of this study is to analyze the influence of EO, environmental factors (dynamism, hostility and complexity) on social impact performance and social causes performance.

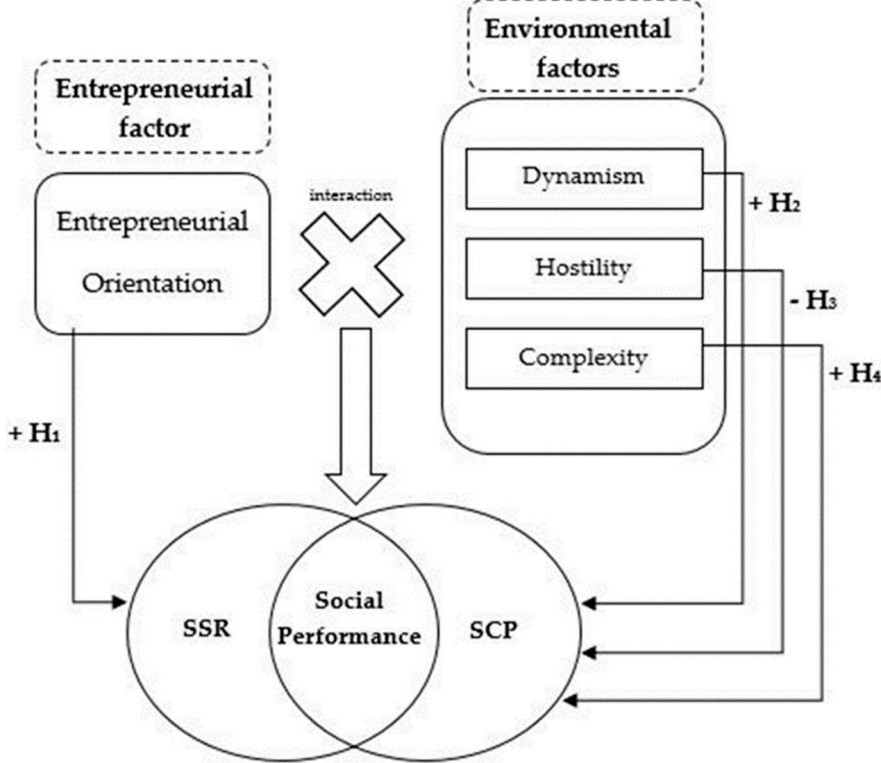

**Figure 1.** Configurational framework.

## 3. Materials and Methods

### 3.1. Data and Data Analysis

The data regarding entrepreneurial orientation, environmental factors and social performance of the sports clubs were obtained through a questionnaire sent online through the University of Valencia's own platform (Limesurvey 2.5) to directors and technical, economic and sports managers of Spanish sports clubs. The sample was collected during the months of October to December 2019. After sending the questionnaire to 1368 sports clubs, a total of 224 questionnaires were obtained, of which 15 were eliminated because they were incomplete. A total of 209 were considered for this study. This represents an effective response rate of 16.37%, with the final useful rate being 15.28%. The details of the sample are shown in Table 1.

**Table 1.** Sample distribution (*n* = 209).

| Items | Freq | Percent | Items | Freq | Percent | Items | Freq | Percent |
|---|---|---|---|---|---|---|---|---|
| **Competition Level** | | | **Seniority (Years)** | | | **Funding** | | |
| Loc-Aut | 69 | 33% | <15 | 68 | 32.5% | >public | 25 | 12% |
| Nat-Int | 140 | 67% | 16–30 | 56 | 26.8% | Public-private | 11 | 5.3% |
| | | | >31 | 85 | 40.7% | >private | 173 | 82.8% |

Note: Freq = frequency.

Of the 209 sports clubs analyzed, 33% are local-autonomic category sports clubs (*n* = 69) and 67% are national-international category clubs (*n* = 140). 32.5% (*n* = 68) are less than 15 years old, 26.8% (*n* = 56) between 16 and 30 years old and 40.7% more than 31 years old (*n* = 85). The number of workers ranges from 0 to 100, with an average of 5.62 (SD = 11.05) and a range of volunteers from 0 to 80, with an average of 12.87 (SD = 13.77). Similarly, the range of athletes is between 2 and 2000, with an average of 196.49 (SD = 230.85). Taking into account the type of financing, 12% (*n* = 25) are a

majority public financing, 5.3% ($n = 11$) have equal financing, while 82.8% ($n = 173$) are a majority private financing.

The analyses were performed from a symmetric approach with a hierarchical regression model (HRM) that identifies symmetric relationships of one or several independent variables on a dependent variable, and from an asymmetric approach using qualitative comparative analysis (QCA) that allows us to identify an asymmetry in the existing relationships. Both methodologies are complementary and help to understand in more detail the entrepreneurial and environmental behavior of sports clubs. The analysis through HRM was carried out with IBM SPSS Statistics 24 software in two steps: in the first step the entrepreneurial factor (EO) was introduced and in the second step the environmental factors (dynamism, hostility and complexity) were introduced.

In addition, to carry out the analysis using QCA, the participants' response for EO, dynamism, hostility, complexity, SIP and SCP was calibrated. Following Prado-Gascó and Calabuig-Moreno [72], to recalibrate the values, we must first multiply the items that compose it and consider the three thresholds [73]: 90% (completely in the set), 50% (intermediate agreement level) and 10% (out of the set) as previously proposed by these authors [74].

This was followed by a needs analysis to find out if any of the conditions were necessary, and a sufficiency analysis carried out in two separate stages [75]. The first stage analyzes a truth table showing all possible combinations of causal conditions, while in the second stage QCA develops three possible solutions: parmonious solution, complex solution and intermediate solution. In the present study, the intermediate solution is represented as recommended [76].

### 3.2. Measures of Variables

The measurement scales used in the questionnaire were adapted from previous studies, while the social performance scale was created and validated by us. The instrument used to measure entrepreneurial orientation was created by Covin and Slevin [26] and later adapted by Engelen et al. [30]. It consists of eight items distributed in three dimensions: risk-taking, innovation and proactivity. The scale as a whole presents good psychometric properties ($\alpha = 0.78$), just as it has presented good psychometric properties in previous studies [5]. The scale created by Miller et al. [58] and adapted by Zahra [59] has been used to measure the environment. However, this last scale has been adapted by us to the associative sport context, presenting good psychometric properties: the general Cronbach alpha of the scale was 0.76. The scale includes the dimensions of dynamism ($\alpha = 0.79$), hostility ($\alpha = 0.70$) and complexity ($\alpha = 0.91$). Finally, to measure the social performance of a non-profit sports organization, a two-factor scale was created, since the previous literature does not provide a similar scale in the sports association field. This scale has good psychometric properties: the general Cronbach alpha of the scale was 0.75. The scale encompasses the social impact performance (SIP) factors ($\alpha = 0.73$) with three items (e.g., "We have increased the number of programs and/or services we offer") and social causes performance (SCP) ($\alpha = 0.78$) (e.g., "Our organization has a social program to help groups with special needs (people with low incomes, people with disabilities, migrants, etc.)"). All items were answered on a Likert scale with a range from 1 ("strongly disagree") to 7 ("strongly agree").

### 3.3. Ethical Considerations

This project has been carried out under the supervision of the Ethics Commission for Experimental Research (1119833). All the participants in the study were informed of the purpose and the procedure that we would follow, as well as the anonymity and confidentiality of the data collected.

### 4. Results

In this section, the quantitative data collected, interpreted and presented in tables were analyzed.

*4.1. Hierarchical Regression Model*

The predictive powers of the variables studied in the study were analyzed using the hierarchical regression model (HRM): the dependent variables were SIP and SCP. Predictive variables were analyzed in two steps. First step: entrepreneurial variables (entrepreneurial orientation), second step: environmental variables (dynamism, hostility and complexity).

In the case of SIP, the total predictive value was 14% ($\Delta R2 = 0.14$; $p = 0.001$). Taking into account the inclusion by steps: in the first step, EO explains 12% ($\Delta R2 = 0.12$; $p = 0.001$), and in the second step, by including the environment variables (dynamism, hostility and complexity), 14% is explained ($\Delta R2 = 0.14$; $p = 0.001$), there being a significant change in the predictive power by including the environment variables ($\Delta R2 = 0.034$; $p = 0.03$). Considering all the variables studied, EO ($\beta = 0.33$; $p = 0.001$) and complexity ($\beta = 0.15$; $p = 0.02$) were significant predictors in the last step for SIP.

In the same line, in the case of SCP, the total predictive value was 13% ($\Delta R2 = 0.13$; $p = 0.001$). In the first step, EO explained 7% ($\Delta R2 = 0.07$; $p = 0.001$); however, after including in the second step the variables of the environment (dynamism, hostility and complexity), 13% was explained ($\Delta R2 = 0.13$; $p = 0.001$). Therefore, the change after including the environment variables was significant ($\Delta R2 = 0.07$; $p = 0.001$). Taking into account all the variables studied, EO ($\beta = 0.24$; $p = 0.001$), dynamism ($\beta = 0.16$; $p = 0.02$) and complexity ($\beta = 0.15$; $p = 0.02$) were the significant predictors of SCP (Table 2).

**Table 2.** Hierarchical regressions for the entrepreneurial factor (EO) and environmental factor on of social impact performance (SIP) and social causes performance (SCP).

| Variables | SIP | | SCP | |
|---|---|---|---|---|
| **Predictors** | $R^2$ | $\beta$ | $R^2$ | $\beta$ |
| Step 1 | 0.12 *** | | 0.07 *** | |
| EO | | 0.35 *** | | 0.27 *** |
| Step 2 | 0.03 * | | 0.07 *** | |
| EO | | 0.33 ** | | 0.24 *** |
| Dynamism | | 0.07 | | 0.16 * |
| Hostility | | −0.01 | | 0.06 |
| Complexity | | 0.15 * | | 0.15 * |
| Total $R^2$adj | 0.14 *** | | 0.13 *** | |

Note: * $p \leq 0.05$; ** $p \leq 0.01$; *** $p \leq 0.001$.

*4.2. Comparative Qualitative Analysis of Fuzzy Sets (fsQCA)*

First, the main statistical descriptors (mean, standard deviation, minimum and maximum) and the calibration values of all the variables were studied to convert them into variables into fuzzy-set conditions. The percentiles presented belong to the 10th percentile, 50th percentile (median) and 90th percentile (Table 3).

After conducting the necessity analysis, no causal condition was necessary for the presence or absence (marked by ~) of any of the outcomes, since none of the cases analyzed had a consistency greater than 0.90, as stated by Ragin [76] (Table 4).

Finally, the sufficiency test was performed and the threshold was set at a significant break in the distribution of the consistency results set out in the truth table [77]. Our overall consistency scores range from 0.76 to 0.78, which is acceptable according to Ragin [76], who recommends a minimum consistency threshold of 0.75. The frequency cut-off for all models analyzed was set at 1.

**Table 3.** Main descriptions and calibrations.

|  |  | EO | DIN | HOS | COM | SIP | SCP |
|---|---|---|---|---|---|---|---|
| N | Valid | 209 | 209 | 209 | 209 | 209 | 209 |
|  | Missing | 0 | 0 | 0 | 0 | 0 | 0 |
| Mean |  | 973,838.97 | 90.90 | 563.64 | 76.37 | 216.84 | 28.33 |
| SD |  | 100,787.63 | 79.62 | 523.96 | 89.77 | 98.38 | 15.99 |
| Min |  | 50 | 1 | 4 | 1 | 8 | 1 |
| Max |  | 5,764,801 | 343 | 2,401 | 343 | 343 | 49 |
| Calibrated values |  |  |  |  |  |  |  |
|  | 10 | 36,000 | 4 | 630 | 1 | 64 | 5 |
| Percentiles | 50 | 735,000 | 75 | 8,000 | 45 | 216 | 28 |
|  | 90 | 2,286,144 | 196 | 38,880 | 216 | 343 | 49 |

Note: SD = standard deviation; EO = Entrepreneurial Orientation; DIN = dynamism; HOS = hostility; COM = complexity.

**Table 4.** Necessary conditions.

|  | SIP | | ~SIP | | SCP | | ~SCP | |
|---|---|---|---|---|---|---|---|---|
|  | **Cons** | **Cov** | **Cons** | **Cov** | **Cons** | **Cov** | **Cons** | **Cov** |
| EO | 0.64 | 0.72 | 0.52 | 0.53 | 0.62 | 0.69 | 0.53 | 0.55 |
| ~EO | 0.58 | 0.57 | 0.73 | 0.65 | 0.59 | 0.58 | 0.70 | 0.63 |
| Dynamism | 0.61 | 0.67 | 0.55 | 0.54 | 0.62 | 0.67 | 0.54 | 0.54 |
| ~Dynamism | 0.58 | 0.59 | 0.66 | 0.61 | 0.57 | 0.57 | 0.67 | 0.62 |
| Hostility | 0.57 | 0.68 | 0.56 | 0.59 | 0.63 | 0.67 | 0.70 | 0.58 |
| ~Hostility | 0.64 | 0.61 | 0.67 | 0.60 | 0.63 | 0.63 | 0.65 | 0.67 |
| Complexity | 0.59 | 0.70 | 0.47 | 0.51 | 0.60 | 0.70 | 0.47 | 0.51 |
| ~Complexity | 0.58 | 0.55 | 0.72 | 0.62 | 0.58 | 0.55 | 0.72 | 0.62 |

Note: Cons = consistency; cov = coverage.

On the one hand, in relation to the prediction for high levels of SIP, four paths or combinations explained 62% of the cases with high levels of social performance (overall consistency = 0.76; overall coverage = 0.62). The most important path to predict SIP was the result of the "EO*DIN" interaction (raw coverage = 0.45; unique coverage = 0.03; consistency = 0.82), followed by the "DIN*~HOS*COM" interaction (raw coverage = 0.43; unique coverage = 0.05; consistency = 0.80) and, the third most important was the combination of "EO*COM" (raw coverage = 0.41; unique coverage = 0.04; consistency = 0.84). Therefore, in relation to the prediction for low SIP levels (~SIP), two paths explain 48% (overall consistency = 0.76; overall coverage = 0.48). The most important combination is "~EO*~HOS*~COM" (raw coverage = 0.41; unique coverage = 0.08; consistency = 0.77), followed by the interaction "~EO*~DIN*~COM" (raw coverage = 0.41; unique coverage = 0.07; consistency = 0.78). All of them are adequate since, following the proposal of Eng and Woodside [66], the raw coverage is between 0.25 and 0.65.

On the other hand, for the prediction for high levels of SCP there are three paths that explain 56% of the total cases (overall consistency = 0.78; overall coverage = 0.56). The most important path is the result of the combination of "EO*DIN" (raw coverage = 0.40; unique coverage = 0.12; consistency = 0.78), the second most important path is the interaction of "EO*COM" (raw coverage = 0.40; unique coverage = 0.04; consistency = 0.82), and, finally, the third path is the result of the combination of "DIN*COM" (raw coverage = 0.39; unique coverage = 0.10; consistency = 0.79). However, analyzing the explanation for low levels of CPS (~SCP), two paths explain 47% of the total number of existing cases (overall consistency = 0.76; overall coverage = 0.47). First, the most important path is the result of the combination "~EO*HOS*~COM" (raw coverage = 0.41; unique coverage = 0.08; consistency = 0.78), and the second most important solution is the one formed by the interaction "~EO*~DIN*~COM" (raw coverage = 0.40; unique coverage = 0.06; consistency = 0.77) (Table 5).

**Table 5.** Summary of three main sufficient conditions of RS and RCS (~) (Intermediate solution).

| Frequency Cut-Off: 1 | SIP | | | ~SIP | | | SCP | | | ~SCP | | |
|---|---|---|---|---|---|---|---|---|---|---|---|---|
| | Cons Cut: 0.80 | | | Cons Cut: 0.80 | | | Cons Cut: 0.81 | | | Cons Cut: 0.81 | | |
| | 1 | 2 | 3 | 1 | 2 | 3 | 1 | 2 | 3 | 1 | 2 | 3 |
| EO | ● | | ● | ○ | ○ | - | ● | ● | | ○ | ○ | - |
| Dynamism | ● | ● | | | ○ | - | ● | | ● | | ○ | - |
| Hostility | | ○ | | ○ | | - | | | | ● | | - |
| Complexity | | ● | ● | ○ | ○ | - | | ● | ● | ○ | ○ | - |
| Raw coverage | 0.45 | 0.43 | 0.41 | 0.41 | 0.41 | - | 0.43 | 0.40 | 0.39 | 0.41 | 0.40 | - |
| Unique Coverage | 0.03 | 0.05 | 0.04 | 0.08 | 0.07 | - | 0.12 | 0.04 | 0.10 | 0.08 | 0.06 | - |
| Consistency | 0.82 | 0.80 | 0.84 | 0.77 | 0.78 | - | 0.78 | 0.82 | 0.79 | 0.78 | 0.77 | - |
| **Overall solution Consistency** | | 0.76 | | | 0.76 | | | 0.78 | | | 0.76 | |
| **Overall Solution Coverage** | | 0.62 | | | 0.48 | | | 0.56 | | | 0.47 | |

● = presence of condition (High levels or positive levels), ○ = absence of condition, ~ = absence of condition (Low levels). All sufficient conditions are adequate, raw coverage between 0.39 and 0.45. Expected address vector for SIP, SCP: 1,1,0,1 (0: absent; 1: present) [78]. Expected address vector for ~SIP, ~SCP: 0,0,1,0.

## 5. Discussion

Given the importance of entrepreneurship for researchers, policy makers and organizational managers, the EO-Performance relationship has been widely studied [26,64], with EO being considered one of the most studied constructs in the field of entrepreneurship [34,39]. However, although Covin and Slevin [26] already studied the role of the external environment in the overall performance of the organization, there is no previous literature analyzing the role that the interaction of EO and environmental factors has on social performance in sports clubs. Therefore, the main objective of the study was to understand the influence that entrepreneurial factors (EO) and external environment factors (dynamism, hostility and complexity) have on the social performance of sports clubs, addressing social performance from two perspectives: social impact and attention to social causes.

Based on the results obtained, EO showed a positive prediction on SIP and SCP in the analyses carried out through HRM and appears in four of the six most important paths to high levels of social performance. Therefore, our first hypothesis can be accepted. This is linked by the above-mentioned authors, who have analyzed the influence of EO on the performance of sports organizations [24,41], however, the previous authors analyze economic and sports performance and economic performance, respectively. However, previous studies have indeed corroborated the influence of EO on social performance in sports clubs [16]. Sport entrepreneurship can have an important social impact [37], as it promotes the creation of social activities that are a means to assist in the health and well-being of its members or participants [79]. In this context, the term sport-for-development (SFD) becomes relevant [80]. SFD activities encompass activities with an important social influence for the improvement of general aspects such as health, welfare, social inclusion, and even to enhance the social development of communities or countries. Due to their high social impact, sports organizations may be forced to have an entrepreneurial attitude in order to achieve added value in the face of competition and maintain sustainability.

As Ratten [22] (p. 5) states, "in a sport entrepreneurial ecosystem there needs to be many resources that can transform innovative ideas into a reality". Therefore, due to the constant changes, the heterogeneity of clients and users and the continuous appearance of new sports disciplines, sports clubs are forced to have an entrepreneurial attitude in order to succeed in social performance. However, the prediction analyzed in this study has not only assessed the role of EO on social performance, but has also analyzed it in interaction with the external environment of the organizations. In this study, the influence of EO on social performance is positive and significant in itself, but it is considerably enhanced when environmental factors are introduced. Nevertheless, we found some differences depending on the dimension of social performance analyzed. According to the results obtained, dynamism positively predicts SCP, while it is a sufficient factor in interaction with EO

for high levels of SIP and SCP. The same happens with the external environment factor, complexity, which predicts SIP and SCP according to data obtained through HRM and is present in four of the six combinations for high levels of general social performance. Similarly, both dynamism and complexity are absent in most of the low level SIP and SCP pathways. Therefore, as stated above, hypothesis 2 and 3 are accepted.

These data fit with the idea that environmental factors such as dynamism can influence organizational performance [63,81], just as the search for opportunities and innovative actions are more important in changing and uncertain environments than in predictable or calm conditions [60]. According to Lumpkin and Dess [71], dynamism is a moderating factor in the EO-Performance relationship, being the major entrepreneurial attitude in dynamic environments. These analyses have been carried out in medium or small for-profit organizations; however, this relationship has not been widely studied in the associative sports field. Thus, this study offers innovative information on the influence of entrepreneurial and environmental factors on social performance.

Finally, hostility is the external environment factor that has generated most controversy among academics [49,82], as there are disparate ideas when analyzing their relationship or influence on organizational performance. There are academics and practitioners who defend their moderating role or direct influence on organizational performance in a negative way [49,71], while other authors argue, conversely, that this factor has a direct influence or positive moderating role on performance [26,70]. In the present study, hypothesis 4 was exposed as a negative influence on social performance, and according to the results obtained through HRM there is no prediction for SIP or SCP. However, the results obtained through QCA show that low levels of hostility are present in a path to high levels of SIP (along with the presence of dynamism and complexity), and high levels of hostility are present in one of the two paths to low levels of SIP and SCP. Therefore, hypothesis 4 can be accepted. Hostility has a negative influence on interactions towards high levels of social performance, regardless of the dimension that we analyze.

In the present study, analyses have been carried out using two different methodologies. On the one hand, hierarchical regression model (HRM), and, on the other hand, comparative qualitative analysis, with the aim of knowing the contribution of entrepreneurial and environmental factors on the social performance of sports clubs. Comparing the two methodologies used, fsQCA models have a higher predictive value than HRM, since they analyze interactions between the factors for the same result, while HRM only considers direct and linear contributions. This is known as the "principle of equifinality", understood as the possibility of achieving the same results from different combinations of factors. This is very operational and useful from a practical point of view, since there are immovable conditions, but other environmental factors that have not been considered with HRM if they are taken into account with QCA. However, despite the complementarity that QCA has with other existing methodologies, it is generally not used in studies in the field of sports [74].

## 6. Conclusions

The results of this study show a positive influence of EO on SIP and SCP. This prediction is significantly higher when external environmental factors such as dynamism or complexity are involved. Therefore, dynamism and complexity have a positive influence on the social performance of Spanish sports clubs. However, the social performance of these organizations has a negative relationship with the hostility of its environment.

The use of different methodologies allows us to complement the results obtained with HRM. This combination is useful to obtain more information than that provided by traditional linear regressions. QCA considers different combinations of conditions for the same result. EO is a condition that appears in most of the paths to SIP and SCP, and the low levels of EO in all combinations of conditions for low levels of SIP and SCP stand out. Therefore, despite not being a necessary condition, it has a wide positive influence on SIP and SCP. Similarly, dynamism and hostility are sufficient conditions for high levels of SIP and SCP, while hostility has a negative relationship with SIP and SCP.

Undoubtedly, understanding the mechanisms that affect the performance of organizations is already an important practical contribution, since it gives managers and executives a rigorous vision of the factors that contribute to improving their operation. The results obtained reinforce, in the first place, the already proven relationship between EO and performance in organizations, and in this case, in sports clubs. As it happens in the business field, the development and promotion of an entrepreneurial culture in the sports club can contribute to improve its performance from two perspectives: (i) a first one of social impact understood as penetration or outreach in the environment; and (ii) a second one understood as an improvement of their attention to social causes.

Managers and directors can see an opportunity for the development of their image and notoriety in both ways: on the one hand, it can increase their popularity and influence in the environment, and on the other hand it can improve their reputation. The former allows for an increase in the number of athletes, members and possibly income, while the latter can improve the club's image, even beyond the intrinsic value that social initiatives provide. Governments and policy-makers can develop an understanding of which mechanism results and performance can be better leveraged at the social level. Again, both perspectives of social performance are particularly useful: social impact is aligned with the role of sports clubs as public policy partners in promoting and popularizing sport, and social causes performance in also serves a public policy objective of developing social policies for integration and correcting market failures. In all cases, and considering the environmental factors, the evaluation and identification of high levels of EO in this type of organizations would allow governments to make better use of public policies of subsidy and aid, as they would be provided to non-profit organizations with the capacity to make better use of them.

Similarly, knowing the characteristics of the environment in which the sports organization operates is fundamental before making entrepreneurial decisions. This study provides new information about the influence that entrepreneurship and the environment have on one of the most important performances for sports clubs. This can partially help the technical and managerial body of sports clubs to carry out innovative actions and show a more entrepreneurial attitude depending on the context in which the organization is at that moment. It can also help develop educational sports programs with content related to entrepreneurship. Sport entrepreneurship and the context in which organizations operate are important factors to assess from an educational point of view for future managers or directors of sports organizations or sports entrepreneurs.

However, this study has related limitations. It was carried out in the context of Spanish sport, so the geographical component could be a limitation. The questionnaire was sent online and answered by directors or managers of the organization based on subjective data of the organization; therefore, this could be a limitation. As future lines of research, it would be interesting to combine measures of subjective social performance with measures of objective social performance. Finally, it might be interesting to carry out additional qualitative research when interviewing managers and heads of sports clubs to find out more detailed information about entrepreneurship in this type of organization and the influence or lack of influence of the environment on the final performance.

**Author Contributions:** Conceptualization, J.M.N.-P.; methodology, P.E.-F.; software, P.E.-F.; formal analysis, P.E.-F.; investigation, J.M.N.-P.; resources, A.M.G.-T.; writing—original draft preparation, P.E.-F.; supervision, A.M.G.-T.; project administration, J.M.N.-P.; funding acquisition, A.M.G.-T. All authors have read and agreed to the published version of the manuscript.

**Funding:** This research was funded by Generalitat Valenciana, grant number GV/2019/133. The first author of this study received funding from the predoctoral scholarship "ACIF/2017/294" financed by the European Social Fund.

**Acknowledgments:** All authors are grateful for the voluntary collaboration of the sports clubs that participated in this study.

**Conflicts of Interest:** The authors declare no conflict of interest.

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
