# Peer review of "Exploring Environmental and Entrepreneurial Antecedents of Social Performance in Spanish Sports Clubs: A Symmetric and Asymmetric Approach"

_sustainability, doi:10.3390/su12104234_

Round 1

Reviewer 1 Report

The authors undertake a multi-faceted issue that is of importance to scholars, business, sport management, and policy-makers alike. The issues are relevant to the journal audience. Rather than focus on the obvious strengths of the manuscript, I devote the following critique to areas of potential improvement.

My chief concerns are:

#1 Abstract: The data used in the study should be included here.

#2 Introduction: In the introduction, the gaps find out on past literature and how this research tries to fill this need to be explored in a dep way. To say: “the literature is scarce in studies that jointly analyze the influence of environmental factors and entrepreneurial factors on the social performance of sports clubs” is not enough. Why is important to do this study?  

#3 Literature Review: The authors formulate four propositions, however, the study is quantitative and in this sense is more suitable to formulate research hypotheses. The three last “propositions” are all together and it is better to find arguments from the literature to support each one of them separately.

#4 Model Proposal: section 3.1 should appear out of section 3 – Materials and methods. I suggest putting the model at the end of the literature review.

The model (fig.1) should reflect the “propositions”. It is important to look for the model and see the P1, P2, P3, and P4 there.

#5 Data: Section 3.5 Data analysis should be merged with section 3.1 data.

#6 Conclusion: The theoretical and practical contributions and managerial implications need to be highlighted.

I wish the author/s all the best in developing their manuscript.

Reviewer 2 Report

Dear Authors,

I think you found an interesting topic to be studied. Generally speaking I enjoyed reading the manuscript and I have only some suggestions to improve the positioning of the paper. The general logic flow is coherent but I feel like you are missing an important perspective on your issue. In the last years has emerged a specific paradigm in entrepreneurship that is call "sport Entrepreneurship" (Ratten, 2010). As you want to inquire the EO effect on social performance of sport club, this is one of the main research avenue of the field (Pellegrini et al., 2020). Indeed, sport entrepreneurship is often display as an improvement of social community impact (Hemme et al., 2017). This aspect of sport entrepreneurship is never really touched but I strongly believe that it would position in a better way your paper. Thus I suggest to revise the introduction and literature review to full acknowledge the existence of important research streams. Yet, considering the important insights you may grasp from this literature, for a review see Pellegrini et al. 2020 that is a bibliometric study on sport entrepreneurship, your discussion and implications can also be improved.

A minor thing, Maybe you could specify better the dynamism variable. While I do agree that this can stimulate the impact, I would also say that in the long terms this "turbulence" may reduce the ability of a firm (even a social firm and thus sport clubs) (see Ge et al., 2019).

All considered, once you include sport entrepreneurship in your discussion, I guess the paper is a nice piece of knowledge.

References

Ge, J., Xu, H., & Pellegrini, M. M. (2019). The effect of value co-creation on social enterprise growth: Moderating mechanism of environment dynamics. Sustainability11(1), 250.

Hemme, F., Morais, D. G., Bowers, M. T., & Todd, J. S. (2017). Extending sport-based entrepreneurship theory through phenomenological inquiry. Sport Management Review, 20(1), 92–104.

Pellegrini, M. M., Rialti, R., Marzi, G., & Caputo, A. (2020). Sport entrepreneurship: A synthesis of existing literature and future perspectives. International Entrepreneurship and Management Journal, 1-32.

Ratten, V. (2010). Developing a theory of sport-based entrepreneurship. Journal of Management & Organization, 16(4), 557–565.

Round 2

Reviewer 1 Report

Dear Author(s)

I have looked at your efforts to attend to my two comments and am very pleased to congratulate you on the changes you have made to the manuscript.

I am delighted to now recommending an 'Accept' decision.

All the very best in your future writing!

Best Regards